# Evaluation of the predictive ability of ultrasound-based assessment of breast cancer using BI-RADS natural language reporting against commercial transcriptome-based tests

**Neema Jamshidii[1,2]\*, Jason Chang[2], Kyle Mock[3], Brian Nguyen[3], Christine Dauphine[3], Michael D. Kuo[4]\***

**1** UCLA Department of Radiological Sciences, Los Angeles, CA, United States of America, **2** UCLA David Geffen School of Medicine, Los Angeles, CA, United States of America, **3** Harbor-UCLA Medical Center, Department of Surgery, Los Angeles, CA, United States of America, **4** Department of Radiology, The University of Hong Kong, Hong Kong, China

\* mikedkuo@gmail.com (MDK); njamshidi@mednet.ucla.edu (NJ)

**Data Availability Statement:** All relevant data are within the paper and its Supporting Information files.

## Abstract

### Purpose

The objective of this study was to assess the classification capability of Breast Imaging Reporting and Data System (BI-RADS) ultrasound feature descriptors targeting established commercial transcriptomic gene signatures that guide management of breast cancer.

### Materials and methods

This retrospective, single-institution analysis of 219 patients involved two cohorts using one of two FDA approved transcriptome-based tests that were performed as part of the clinical care of breast cancer patients at Harbor-UCLA Medical Center between April 2008 and January 2013. BI-RADS descriptive terminology was collected from the corresponding ultrasound reports for each patient in conjunction with transcriptomic test results. Recursive partitioning and regression trees were used to test and validate classification of the two cohorts.

### Results

The area under the curve (AUC) of the receiver operator curves (ROC) for the regression classifier between the two FDA approved tests and ultrasound features were 0.77 and 0.65, respectively; they employed the 'margins', 'retrotumoral', and 'internal echoes' feature descriptors. Notably, the 'retrotumoral' and mass 'margins' features were used in both classification trees. The identification of sonographic correlates of gene tests provides added value to the ultrasound exam without incurring additional procedures or testing.

**Funding:** The authors received no specific funding for this work.

**Competing interests:** The authors have declared that no competing interests exist.

**Abbreviations:** NLP, natural language processing; BI-RADS, Breast Imaging Reporting and Data System; ANOVA, analysis of variance; AUC, area under the curve; ROC, receiver operator characteristic; CART, curve, classification and regression trees.

## Conclusions

The predictive capability using structured language from diagnostic ultrasound reports (BI-RADS) was moderate for the two tests, and provides added value from ultrasound imaging without incurring any additional costs. Incorporation of additional measures, such as ultrasound contrast enhancement, with validation in larger, prospective studies may further substantiate these results and potentially demonstrate even greater predictive utility.

## Introduction

Breast cancer continues to be a significant problem around the world, accounting for 29% of newly diagnosed cancers in women, with women in the United States having a 12.3% chance of developing breast cancer over their lifetimes [1]. Despite increased prevalence, breast cancer death rates have decreased by 36% from 1989 to 2012 due to screening and improved treatment regimens [2]. The use of ultrasound in screening, particularly in women younger than 45 years of age with fibroglandular, dense tissue, has improved cancer detection sensitivity in comparison to mammography alone [3]. The ability of ultrasound to detect small (< 1 cm), mammographically occult lesions [4] as well as potentially downgrading breast masses [5] provide examples of how ultrasound can further guide clinical decision making in cancer screening and detection.

The clinical utility of diagnostic modalities that provide more than just an assessment of malignant versus benign, but rather prognostic information has driven the development of transcriptome-based tests. The OncotypeDX® (Genomic Health Inc, Redwood City, CA) test calculates a recurrence score on a scale from 0–100 (higher scores reflect higher risks of recurrence) for early-stage hormone receptor-positive breast cancer, in order to assess the potential benefit from chemotherapy after breast cancer surgery (7). This test is currently included in the National Comprehensive Cancer Network (NCCN) and American Society of Clinical Oncology (ASCO) guidelines. The MammaPrint® (Agendia Inc, Irvine, CA) test calculates a binary (high versus low) recurrence risk score based upon a 70-gene expression profile, and also guides chemotherapy treatment decisions [6, 7]. In patients with an OncotypeDX® score less than 11 or a MammaPrint® low-risk score, chemotherapy can be omitted, and anti-estrogen therapy can be administered alone.

The development of the Breast Imaging-Reporting and Data System (BI-RADS) [8] by the American College of Radiology (ACR) to include standardized language and descriptors in the reporting of breast imaging has proven very successful in standardizing communication among radiologists, oncologists, and surgical oncologists in the decision-making and management of breast lesions [9]. Other fields have successfully developed similar approaches in prostate, lung, and liver lesions (PI-RADS, Lung-RADS, LI-RADS) [10–13]). Although BI-RADS has an established role in breast imaging and interdisciplinary communication (e.g. medical oncologists, surgical oncologists, and internists), with the ongoing development of new treatments and and genomic-based tests, we need to be able to derive as much information as possible from clinical imaging measurements [14, 15]. The BI-RADS categorization of sonographic findings has sufficient positive predictive value to be used as a predictor of malignancy [16]. Along a parallel track, correlations between cross-sectional imaging and molecular profiling have identified potential surrogate roles as imaging biomarkers in a variety of diseases, including breast cancer [17–20].

The clinically established use of ultrasound BI-RADS reporting affords an opportunity to assess how much information can be derived from imaging alone, and whether various descriptors may supplement prognostic gene panels. We sought to assess the potential of breast ultrasound feature descriptors to identify cohorts that would or would not benefit from chemotherapy using structured natural language processing (NLP) of BI-RADS terminology targeting established transcriptomic assays (OncotypeDX® and MammaPrint®).

## Materials and methods

This retrospective study received approval with waiver of written patient consent, from the Institutional Review Board at the Los Angeles Biomedical Research Institute at Harbor-UCLA, and is Health Insurance Portability and Accountability Act (HIPAA)-compliant. All data were fully anonymized for subsequent study analyses.

Patient demographics, histological classification of biopsied samples, receptor status (when available), and transcriptomic-test were extracted from the electronic medical record system. Breast cancer patients were included in this study if they had undergone testing with either of the two genomic assays (OncotypeDX® and MammaPrint®) between April 2008 and January 2013, and were divided into two cohorts depending on which assay was performed. Ultrasound reports for each patient at the time of initial cancer diagnosis were stored in a collection of Microsoft Word® documents (.docx).

The diagnostic ultrasound studies, interpreted and reported by fellowship trained, board-certified breast radiologists (each with at least 4 years experience), with strict adherence to BI-RADS standards [8], were parsed with custom scripts focusing on BI-RADS descriptive terminology. Four classes of ultrasound BI-RADS field descriptors were consistently reported across all reports, accounting for the features that were most consistently described: margins, echogenicity, internal echo pattern, and retrotumoral phenomenon [8]. The cumulative collection of these descriptive terms from all reports defined the dictionary of terms.

The BI-RADS ultrasound report files were parsed as regular expressions focusing on the FINDINGS and IMPRESSION sections of the diagnostic reports using Python version 2.7.10 (https://www.python.org/). The FINDINGS were parsed according to size ('Longitudinal', 'Transverse', 'Anteroposterior'), '*margins*', '*echogenicity*', '*internal echo pattern*', '*internal shadowing*', and '*retrotumoral phenomenon*'. The reported BI-RADS score was extracted from the IMPRESSION section of the reports. '*margins*' values were, 'ill-defined', 'irregular', 'smooth', 'lobulated', or 'N/A'. '*echogenicity*' values included, 'hypoechoic', 'isoechoic', 'anechoic'. '*internal echo pattern*' values included, 'homogeneous' and 'heterogeneous'. '*internal shadowing*' values included, 'small ca++', 'large ca++', and 'none'. '*retrotumoral phenomenon*' values included, 'irregular posterior shadowing' and 'posterior shadowing'. Since the sizes of the masses were not consistently measured in all three dimensions for the majority of the cases, tumor size measurement was discarded from subsequent analyses (S1 and S2 Tables).

The ability of the BI-RADS ultrasound features to predict risk score classification by the OncotypeDX® and MammaPrint® transcriptome-based tests was assessed using recursive partitioning and regression trees (CART) using analysis of variance (ANOVA). Training to testing validation sets were split in 3:1 ratios, randomly split with the Mersenne-Twister random number generator (seed = 202). Twenty-fold cross-validation was performed with a minimum of 10 observations per splitting-node and a minimum of 6 observations per terminal node. We defined an area under the curve (AUC) of the receiver operator characteristic (ROC) curve to be at least 0.9 in order to be considered as a candidate classifier to potentially compete with molecular tissue markers and an AUC of 0.6 to be a reportable but not

competitive as an radiogenomic surrogate. Statistical significance for all portions of the study were defined as $p < 0.05$. The analyses were performed using R (https://cran.r-project.org/).

## Results

### Cohort characteristics

In the two cohorts of patients, 149 had undergone testing with the OncotypeDX® assay and 70 had the MammaPrint® assay. Both of the genetic test cohorts showed a significant association with tumor grade (Tables 1 and 2) as determined by ANOVA ($p <0.05$), as expected. There was no significant association with age, race or tumor histology with OncotypeDX® or MammaPrint® classifications (Tables 1 and 2). There were significant negative correlations between ER and PR for OncotypeDX® and MammaPrint® with weaker correlation coefficients for the latter (S3 and S4 Tables), grossly consistent with other published reports [21, 22].

### Ultrasound imaging features

Collectively the 219 sonographically detectable masses characterized according to five semantic features (see Materials and Methods) and were assessed for a possible 144 different classifications of the masses. All BI-RADS scores were 3 or greater, as would be expected, based upon interpretation and biopsy recommendations of BI-RADS [8, 23]. Since the 'echogenicity' of the masses was described as 'hypoechoic' in 217 out of 219 masses (with one described as 'anechoic' and the other as 'isoechoic'), the descriptor was removed from subsequent analyses. Following removal of the echogenicity feature there remained 48 possible unique sonographic classifications of the masses from 4 different features ('margins', 'internal echo pattern', 'internal shadowing', and 'retrotumoral phenomenon').

### Ultrasound BI-RADS classifiers

The CART classification trees alongside their corresponding ROC curves are presented in Figs 1 and 2, with AUCs of 0.77 (OncotypeDX®, Fig 1) and 0.65 (MammaPrint®, Fig 2). Incorporation of tumor grade information into the regression analysis did not improve the predictive value of the classification trees. Mass margins and retrotumoral phenomena appear at the top of the classification tree for both tests. Additionally, for these cohorts, although there were four different possible values for the tumor margin feature, the classification separation boundaries occurred along binary lines (smooth versus non-smooth and smooth/lobulated versus irregular), which is concordant with "benign versus malignant" suspicion in the BI-RADS based assessment [8].

## Discussion

In this study we analyzed the potential for BI-RADS ultrasound descriptors to track the OncotypeDX® or MammaPrint® classifications using NLP in conjunction with classification and regression trees, resulting the identification of three sonographic features ('margins', 'retrotumoral' and 'internal echoes') that may provide non-invasive correlates of the transcriptome profiles. Through the use of specific terminology and a well-defined vocabulary with systematic report recommendations, ultrasound BI-RADS has been an effective mechanism to provide consistent, transparent, and unambiguous recommendations to referring physicians and their patients to interpret the results of breast imaging studies (14). The use of a structured language and well-defined vocabulary is particularly useful since one challenge with respect to quantitative imaging of breast ultrasound is the non-tomographic, operator dependent nature of image acquisition, resulting in variation in acquisition with respect to anatomic planes as well as

**Table 1. Summary statistics for the OncotypeDX® cohort by age (mean +/- sd years), grade (mean +/- sd), race (A: Asian, AA: African American, C: Caucasian, F: Philipina, H: Hispanic, ME: Middle Eastern, -: not documented), and histology (IDC: invasive ductal carcinoma, ILC: invasive lobular carcinoma, IDC/ILC: invasive ductal carcinoma with lobular features).** The bottom row highlights ANOVA p-values. * indicates statistical significance (p <0.05).

|  | Age (years) | Grade* | Race | Histology |
|---|---|---|---|---|
|  | 54.2+/-9.4 | 1.8+/-0.70 | A: 11 | IDC: 56 |
|  |  |  | AA: 13 | ILC: 8 |
|  |  |  | C: 12 | mucinous: 5 |
|  |  |  | F: 3 | other: 1 |
|  |  |  | H: 30 |  |
|  |  |  | ME: 1 |  |
| **p-value** | 0.6 | 0.0000002 | 0.55 | 0.64 |

ultrasound parameters (e.g. different transducer probes, use of harmonics, differences in gain and time gain compensation, focal zones, etc). These sources of variation limit the application of automated or semi-automated quantitative imaging approaches to ultrasound. However, the BI-RADS descriptions of sonographically detectable masses provide an opportunity to use NLP based methods in order to identify features with prognostic and therapeutic implications and correlates with other diagnostic tests, such as the transcriptomic tests evaluated in this study.

The assessments of ultrasound imaging correlates using standardized language and descriptors compared to their relationship to the FDA approved tissue-based transcriptomic tests (OncotypeDX® and MammaPrint®) provide a biological context to interpret the transcriptomic measurements. For example, hypoechoic masses are concerning for malignancy, and it is such a common observation, that there is no further prognostic information to be derived from it, thus although it is an important BI-RADS feature, it is not an important prognostic predictor (since the ROC curves in Figs 1 and 2 implicitly assume *a priori* that the masses are hypoechoic). Conversely the CARTs enable evaluation of multiple features that portend higher or lower risk in which one feature may be more suspicious for malignancy but another feature is not (e.g. irregular margins but no retrotumoral phenomena, Figs 1 and 2).

## Added value without added costs

The search for imaging correlates of transcriptomic tests can be classified to serve as non-invasive 1) alternatives, 2) complementary, or 3) supplementary roles to more invasive, biopsy-dependent tests. Many radiogenomic applications focus on the first point, which is beneficial when an imaging test is less expensive or cheaper than a tissue-based test. For example,

**Table 2. Summary statistics for the MammaPrint® cohort by age (mean +/- sd years), grade (mean +/- sd), race (A: Asian, AA: African American, C: Caucasian, F: Philipina, H: Hispanic, ME: Middle Eastern, -: not documented), and histology (IDC: invasive ductal carcinoma, ILC: invasive lobular carcinoma, IDC/ILC: invasive ductal carcinoma with lobular features).** The bottom row highlights ANOVA p-values. * indicates statistical significance (p <0.05).

|  | Age (years) | Grade* | Race | Histology |
|---|---|---|---|---|
|  | 51.2+/-10.3 | 2.22+/00.68 | A: 15 | IDC: 132 |
|  |  |  | AA: 38 | IDC/ILC: 3 |
|  |  |  | C: 21 | ILC: 11 |
|  |  |  | F: 5 | mucinous: 1 |
|  |  |  | H: 69 | other: 2 |
|  |  |  | -: 1 |  |
| **p-value** | 0.37 | 0.00023 | 0.16 | 0.23 |

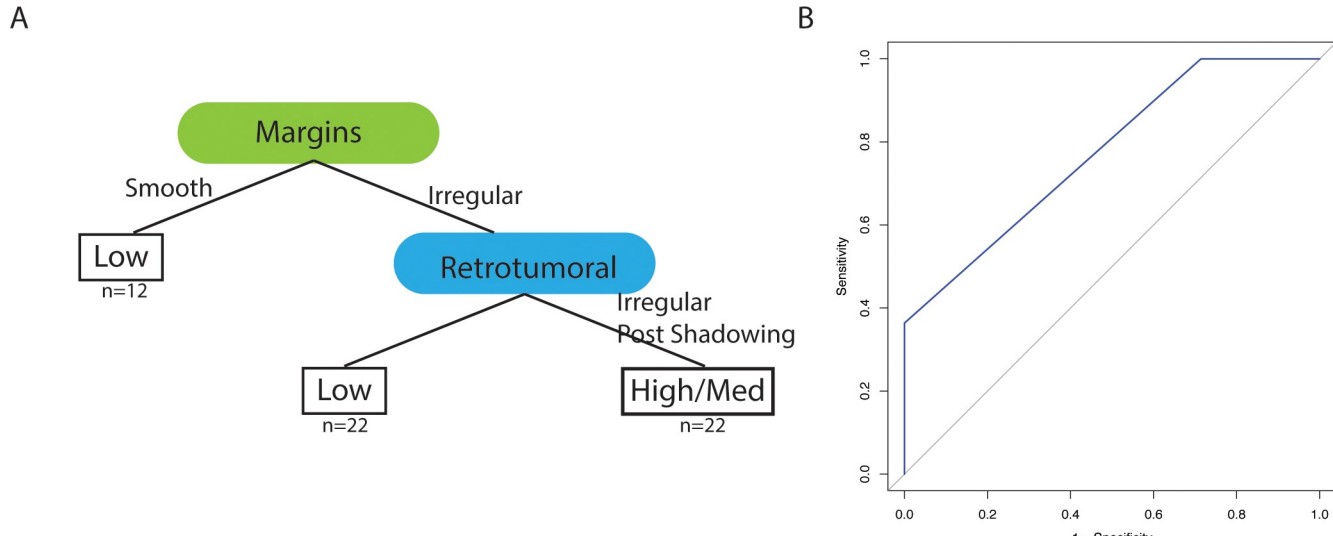

**Fig 1. OncotypeDX® classification based upon BI-RADS ultrasound feature descriptors for hypoechoic breast masses.** A) The classification tree involves two features, the margins of the tumor and the type of shadowing phenomenon for the tumor. B) The area under the ROC curve was 0.77 with 52 subjects in the training set and 18 in the testing set.

recently MRI has been explored as a radiogenomic surrogate for some of these tests. Unfortunately classification of the breast MRI features do not achieve an accuracy that can reasonably compete or provide surrogacy for the established transcriptomic tests [24, 25]. Additionally the cost of MRI scans are non-trivial and rival the MammaPrint® and OncotypeDX® test costs (doubling the cost without providing substantive additional information). However, the ultrasound-based assessments used in this study was focused on the latter the second and third classifications (complementary or supplementary to tissue-based tests).

Although the role of OncotypeDX® and MammaPrint® in management of breast cancer have been promising, there is a non-negligible cost for these tests, in the $3000-$4000 range.

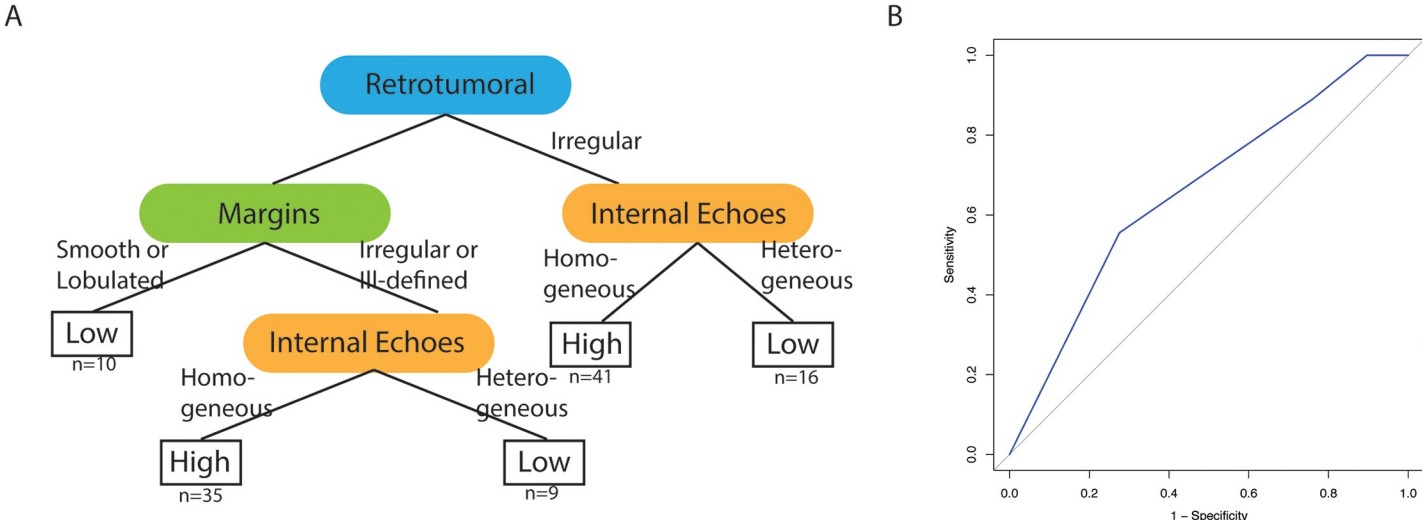

**Fig 2. MammaPrint® classification based upon BI-RADS ultrasound feature descriptors for hypoechoic breast masses.** A) The classification tree involves three BI-RADS ultrasound features, the type of shadowing phenomenon for the tumor, the margins of the tumor, and the internal echo pattern. B) The area under the ROC curve was 0.65 with 111 subjects in the training set and 38 in the testing.

Recent evidence suggests that this may not be cost effective [26], thus it would be beneficial to have a low- or no-cost non-invasive screening test, to determine whether there would be added value from these tests. In a similar vein, the cost of bilateral breast MRI is on average $3000, nearly ten-fold the cost of breast ultrasound [27, 28]. In contrast, part of the established diagnostic evaluation of breast masses involves the breast ultrasound, so there is no additional cost burden. Given the invasive nature of tissue-based tests and the costs associated with tissue biopsies, processing and analysis, in addition to the costs of commercial tests [29], the use of ultrasound imaging information to help identify cases in which transcriptomic tests may alter patient management, provides a potential means to make the transcriptomic tests more cost effective.

The principle limitations of this study include the sample size, the number of available features, and the lack of quantitative measurements. The difference in the sample size between the two cohorts may in part explain why the performance of the OncotypeDX® predictor exceeded the MammaPrint® predictor and highlights the point that, with larger sample sizes and prospective evaluation at different hospitals, classification performance may improve. Although both tests provide guidance for treatment (i.e. low scores for both tests can justify sole anti-estrogen treatment), the MammaPrint® test is applicable to estrogen receptor positive and negative women, whereas the Oncotype® test has been applied demonstrated to estrogen positive cohorts. The difference in the clinical applications may also provide an explanation for the difference in the performance of the ultrasound feature descriptors. For example, 'internal echoes' may not have any predictive significance in estrogen positive women, although testing this in an independent cohort is warranted before drawing such a conclusion.

Despite the aforementioned limitations, the AUC of the ROC curves for the regression decision trees suggest that there is a role for the use of ultrasound BI-RADS descriptors beyond just a probability assessment for malignancy versus benignity. Incorporation of additional features such as color Doppler flow and ultrasound contrast agents may also further improve the molecular predictive value of breast mass sonography. Furthermore, new contrast agents [30] may provide further improvements in the specificity of the classifier, providing more precise diagnostic and prognostic value. Future studies may also evaluate other genomic tests, such as the Breast Cancer Index, EndoPredict, Mammostrat, and Prosigna Breast Cancer Prognostic Gene Signature Assay, for any potential correlation with imaging studies as well [31–34].

## Conclusions

Although BI-RADS was developed to guide decision making in breast imaging studies and to assess the probability of malignancy, the use of a standardized lexicon and descriptive features for ultrasound masses provided the opportunity to use NLP to construct regression trees classifiers for prognostic FDA approved transcriptome-based tissue tests. Using the structured language of ultrasound BI-RADS, we assessed the ability of ultrasound feature characteristics to predict OncotypeDX® and MammaPrint® transcriptome-based classifications across 219 patients. Interestingly, NLP classifications of the BI-RADS reports were able to generate classification trees that were concordant with the transcriptomic tests. Ultrasound findings, notably the 'retrotumoral' and 'margins' features, if abnormal, may help provide justification to obtain one of the transcriptomic tests; future multi-institutional prospective studies will be important in determining if these observations persist in larger cohorts.

## Supporting information

**S1 Table. Table of imaging features, IHC, race, age, and OncotypeDX® scores.**
(XLSX)

**S2 Table. Table of imaging features, IHC, race, age, and MammaPrint® scores.** (XLSX)

**S3 Table. Summary of correlation between OncotypeDX® and IHC markers.** (XLSX)

**S4 Table. Summary of correlation between MammaPrint® and IHC markers.** (XLSX)

## Author Contributions

**Conceptualization:** Michael D. Kuo.

**Data curation:** Neema Jamshidii, Kyle Mock, Brian Nguyen.

**Formal analysis:** Neema Jamshidii.

**Investigation:** Jason Chang, Christine Dauphine, Michael D. Kuo.

**Methodology:** Neema Jamshidii.

**Resources:** Michael D. Kuo.

**Supervision:** Christine Dauphine, Michael D. Kuo.

**Validation:** Neema Jamshidii.

**Writing – original draft:** Neema Jamshidii.

**Writing – review & editing:** Jason Chang, Christine Dauphine, Michael D. Kuo.

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
