## [Decision Letter · Decision Letter 0]

21 Jun 2019

PONE-D-19-15654

Evaluation of the predictive ability of ultrasound-based assessment of breast cancer using BI-RADS natural language reporting against commercial transcriptome-based tests

PLOS ONE

Dear Dr. Jamshidi,

Thank you for submitting your manuscript to PLOS ONE. After careful consideration, we feel that it has merit but does not fully meet PLOS ONE’s publication criteria as it currently stands. Therefore, we invite you to submit a revised version of the manuscript that addresses the points raised during the review process.

We would appreciate receiving your revised manuscript by Aug 05 2019 11:59PM. To enhance the reproducibility of your results, we recommend that if applicable you deposit your laboratory protocols in protocols.io, where a protocol can be assigned its own identifier (DOI) such that it can be cited independently in the future. For instructions see: http://journals.plos.org/plosone/s/submission-guidelines#loc-laboratory-protocols

We look forward to receiving your revised manuscript.

Kind regards,

Azra Alizad, MD

Academic Editor

PLOS ONE

Journal Requirements:

2.  In your ethics statement in the manuscript and in the online submission form, please provide additional information about the patient records used in your retrospective study. Specifically, please ensure that you have discussed whether all data were fully anonymized before you accessed them and/or whether the IRB or ethics committee waived the requirement for informed consent. If patients provided informed written consent to have data from their medical records used in research, please include this information.

3.

We note that you have indicated that data from this study are available upon request. PLOS only allows data to be available upon request if there are legal or ethical restrictions on sharing data publicly. For information on unacceptable data access restrictions, please see http://journals.plos.org/plosone/s/data-availability#loc-unacceptable-data-access-restrictions.

Additional Editor Comments (if provided):

Thank you for sending your great work to PLOS ONE. Please respond to the reviewers’ comments and submit your revisions to PLOSONE.

Reviewers' comments:

Reviewer's Responses to Questions

**Comments to the Author**

1. Is the manuscript technically sound, and do the data support the conclusions?

Reviewer #1: Yes

Reviewer #2: Partly

2. Has the statistical analysis been performed appropriately and rigorously? 

Reviewer #1: Yes

Reviewer #2: Yes

3. Have the authors made all data underlying the findings in their manuscript fully available?

Reviewer #1: Yes

Reviewer #2: Yes

4. Is the manuscript presented in an intelligible fashion and written in standard English?

Reviewer #1: Yes

Reviewer #2: Yes

5. Review Comments to the Author

Reviewer #1: Summary:

This is an interesting paper that explores an aspect of radiogenomics. I think it certainly adds to the current literature on this subject.

Introduction:

1)Consider changing the first paragraph to focus more on the value of US in breast cancer. It is too generic for the focus of this paper.

Methods:

2) It would be important to know what reporting system is used for US …is it powerscribe where the radiologist can describe any terminology they want (compliant with BIRADS or not) or is it a program like MagView where the radiologist is somewhat forced to use BIRADS terms?

3) Need to mention how many radiologists reported on US and what was their experience in terms of years.

4) 4th paragraph of methods: capitalize where appropriate

Results:

Easy to follow

Discussion

5) I am not sure the entire discussion on costs is applicable. It is ok to mention it in a few sentences but such a long section seems out of scope for this paper.

Reviewer #2: The premise of this study is very intriguing and I would recommend publication after consideration of minor revisions. It is clear that while this is an interesting study with potentially very interesting implications, substantially more data from a very large, multicenter, prospective trial, will be needed to be confident that an AUC of 0.6 truly provides sufficient accuracy for ultrasound features to be considered a supportive adjunct to guide the decision of whether molecular based tissue testing is warranted. The transcriptome tests guide critical clinical decisions regarding the benefit of administering vs. not administering chemotherapy and stratify high risk patients from lower risk patients. A significantly greater level of confidence from a much larger study(ies) will be needed before imaging characteristics of tumors can influence clinical decisions about whether to use a tissue-based tests to clarify the need for chemotherapy. While the authors do not purport to draw a definite conclusion, more definitive language from the authors about the limitations of this study, which in essence provides preliminary data, and the need for additional robust data is strongly recommended.

6. PLOS authors have the option to publish the peer review history of their article (what does this mean?). If published, this will include your full peer review and any attached files.

Reviewer #1: No

Reviewer #2: No

---

## [Author Response · Author response to Decision Letter 0]

5 Aug 2019

We thank the reviewers for their evaluation of the manuscript and constructive comments. The manuscript has been revised to reflect the changes that are described below. 

Reviewer #1: Summary:

This is an interesting paper that explores an aspect of radiogenomics. I think it certainly adds to the current literature on this subject.

Thank you, we too believe that this initial evaluation may prompt further studies to extract further information from ultrasound studies, with the potential to serve as useful adjuncts in motivating the need for tissue based transcriptome studies, in the appropriate cohorts.

Introduction:

1) Consider changing the first paragraph to focus more on the value of US in breast cancer. It is too generic for the focus of this paper.

This is a great point, since the focus of the manuscript is on ultrasound. We have added some additional references and statements highlighting the role and relevance of ultrasound.

Methods:

2) It would be important to know what reporting system is used for US …is it powerscribe where the radiologist can describe any terminology they want (compliant with BIRADS or not) or is it a program like MagView where the radiologist is somewhat forced to use BIRADS terms?

There were template reports, however a dictation system was used (Powerscribe, Nuance). All of the reports used BI-RADS terminology (as provided in the Supporting Information). 

3) Need to mention how many radiologists reported on US and what was their experience in terms of years.

The studies were read by three sub-specialty trained radiologists (all with at least 4 years of experience). Methods were updated to reflect this.

4) 4th paragraph of methods: capitalize where appropriate

Done.

Discussion

5) I am not sure the entire discussion on costs is applicable. It is ok to mention it in a few sentences but such a long section seems out of scope for this paper.

We have abbreviated the paragraph regarding the cost benefits and trade-offs.

Reviewer #2: The premise of this study is very intriguing and I would recommend publication after consideration of minor revisions. It is clear that while this is an interesting study with potentially very interesting implications, substantially more data from a very large, multicenter, prospective trial, will be needed to be confident that an AUC of 0.6 truly provides sufficient accuracy for ultrasound features to be considered a supportive adjunct to guide the decision of whether molecular based tissue testing is warranted. The transcriptome tests guide critical clinical decisions regarding the benefit of administering vs. not administering chemotherapy and stratify high risk patients from lower risk patients. A significantly greater level of confidence from a much larger study(ies) will be needed before imaging characteristics of tumors can influence clinical decisions about whether to use a tissue-based tests to clarify the need for chemotherapy. While the authors do not purport to draw a definite conclusion, more definitive language from the authors about the limitations of this study, which in essence provides preliminary data, and the need for additional robust data is strongly recommended.

Thank you for your comments and review of the manuscript. 

We have added additional text to highlight the fact that additional studies with larger cohorts will be needed to further test the potential utility of these features (particularly multi-institutional, given the potential for imager-dependent variability in ultrasound exams). 

We feel it is worth mentioning however, given that the transcriptomic tests are dependent on insurance company approval and the ability of a patient to have the test is driven in large part by whether they have “good” or “bad” insurance, we feel that any additional data (especially if there is no additional cost associate with it) that can help indicate potential benefit from the transcriptomic test provides some value (especially considering some published studies promoting MR based “radiogenomic” predictors compared to transcriptomic tests have argued for the role of surrogacy with AUCs < 0.8 – to that end, we feel that an AUC ~ 0.7 for an adjunctive test is reasonable).

---

## [Decision Letter · Decision Letter 1]

21 Oct 2019

PONE-D-19-15654R1

Evaluation of the predictive ability of ultrasound-based assessment of breast cancer using BI-RADS natural language reporting against commercial transcriptome-based tests

PLOS ONE

Dear Dr. Jamshidi,

Thank you for submitting your manuscript to PLOS ONE. After careful consideration, we feel that it has merit but does not fully meet PLOS ONE’s publication criteria as it currently stands. Therefore, we invite you to submit a revised version of the manuscript that addresses the points raised during the review process.

We would appreciate receiving your revised manuscript by Dec 05 2019 11:59PM. To enhance the reproducibility of your results, we recommend that if applicable you deposit your laboratory protocols in protocols.io, where a protocol can be assigned its own identifier (DOI) such that it can be cited independently in the future. For instructions see: http://journals.plos.org/plosone/s/submission-guidelines#loc-laboratory-protocols

We look forward to receiving your revised manuscript.

Kind regards,

Azra Alizad, MD

Academic Editor

PLOS ONE

Additional Editor Comments (if provided):

Thank you very much for your submission and your response to the reviewers' comments.

Reviewers' comments:

Reviewer's Responses to Questions

**Comments to the Author**

1. If the authors have adequately addressed your comments raised in a previous round of review and you feel that this manuscript is now acceptable for publication, you may indicate that here to bypass the “Comments to the Author” section, enter your conflict of interest statement in the “Confidential to Editor” section, and submit your "Accept" recommendation.

Reviewer #1: All comments have been addressed

Reviewer #3: (No Response)

2. Is the manuscript technically sound, and do the data support the conclusions?

Reviewer #1: Yes

Reviewer #3: Yes

3. Has the statistical analysis been performed appropriately and rigorously? 

Reviewer #1: Yes

Reviewer #3: I Don't Know

4. Have the authors made all data underlying the findings in their manuscript fully available?

Reviewer #1: Yes

Reviewer #3: Yes

5. Is the manuscript presented in an intelligible fashion and written in standard English?

Reviewer #1: Yes

Reviewer #3: Yes

6. Review Comments to the Author

Reviewer #1: Thanks for addressing all the comments. No further comments. I think this will be a valuable manuscript.

Reviewer #3: This is an interesting look at the breast US features, as they correlate with breast cancer prognostic and predictive markers. It is nicely presented. However, I do take issue with the determination, under the method section, that " AUC of 0.6 to be considered as a supportive adjunct to help guide the decision of whether a molecular based tissue test is warranted"(page 37). I am not sure how this conclusion was reached, since this is not a study assessing the need for employing a prognostic marker, but rather a study looking at correlating a prognostic marker with imaging features. The decision on when to use a prognostic marker resides with the oncologist and is influenced by many patient and tumor characteristics and I do not believe it can be made on imaging features alone, at this point in time.

The conclusion that the US features employed here can be used to predict when to use a prognostic marker is, therefore, far reaching and should be revised.

What this study demonstrates is a correlation between the US features and Oncotype DX and, to a lesser degree, to the MammaPrint, but the correlation is moderate at best. Therefore, it can not be used in place of these tests to predict who needs chemotherapy.

The study is promising in that it shows a correlation and I agree with the conclusion that adding other imaging features to the ones evaluated in this study may improve the AUC and hopefully achieve the >0.9 AUC.

The Discussion is very lengthy, as mentioned by another reviewer. I would consider moving some of the discussion under the introduction part, for example the costs of MRI and prognostic markers and focus the discussion on the findings of this study and how it compares to other studies, like you did under the Conclusion section.

7. PLOS authors have the option to publish the peer review history of their article (what does this mean?). If published, this will include your full peer review and any attached files.

Reviewer #1: No

Reviewer #3: No

---

## [Author Response · Author response to Decision Letter 1]

6 Nov 2019

Thank you for the comments. We have enclosed a pdf with our point by point responses.

---

## [Decision Letter · Decision Letter 2]

4 Dec 2019

Evaluation of the predictive ability of ultrasound-based assessment of breast cancer using BI-RADS natural language reporting against commercial transcriptome-based tests

PONE-D-19-15654R2

Dear Dr. Jamshidi,

We are pleased to inform you that your manuscript has been judged scientifically suitable for publication and will be formally accepted for publication once it complies with all outstanding technical requirements.

With kind regards,

Azra Alizad, MD

Academic Editor

PLOS ONE

Additional Editor Comments (optional):

Thank you for submitting your interesting work to PLos One. I gladly recommend your paper for publication.

Reviewers' comments:

Reviewer's Responses to Questions

**Comments to the Author**

1. If the authors have adequately addressed your comments raised in a previous round of review and you feel that this manuscript is now acceptable for publication, you may indicate that here to bypass the “Comments to the Author” section, enter your conflict of interest statement in the “Confidential to Editor” section, and submit your "Accept" recommendation.

Reviewer #3: All comments have been addressed

2. Is the manuscript technically sound, and do the data support the conclusions?

Reviewer #3: Yes

3. Has the statistical analysis been performed appropriately and rigorously? 

Reviewer #3: I Don't Know

4. Have the authors made all data underlying the findings in their manuscript fully available?

Reviewer #3: Yes

5. Is the manuscript presented in an intelligible fashion and written in standard English?

Reviewer #3: Yes

6. Review Comments to the Author

Reviewer #3: I have no additional comments or concerns, and I am satisfied with the response to my previous comments.

7. PLOS authors have the option to publish the peer review history of their article (what does this mean?). If published, this will include your full peer review and any attached files.

Reviewer #3: No

---

## [Editor Report · Acceptance letter]

12 Dec 2019

PONE-D-19-15654R2 

Evaluation of the predictive ability of ultrasound-based assessment of breast cancer using BI-RADS natural language reporting against commercial transcriptome-based tests 

Dear Dr. Jamshidi:

I am pleased to inform you that your manuscript has been deemed suitable for publication in PLOS ONE. Congratulations! Your manuscript is now with our production department. 

With kind regards,

on behalf of

Dr. Azra Alizad 

Academic Editor

PLOS ONE